# Prediction of Metal Additively Manufactured Surface Roughness Using Deep Neural Network

**DOI:** 10.3390/s22207955

**Published:** 2022-10-19

**Authors:** Min Seop So, Gi Jeong Seo, Duck Bong Kim, Jong-Ho Shin

**Affiliations:** 1Department of Industrial Engineering, Chosun University, Gwangju 61452, Korea; 2Department of Manufacturing and Engineering Technology, Tennessee Tech University, Cookeville, TN 38505, USA

**Keywords:** wire + arc additive manufacturing, surface roughness, deep neural network, arc welding

## Abstract

In recent years, manufacturing industries (e.g., medical, aerospace, and automobile) have been changing their manufacturing process to small-quantity batch production to flexibly cope with fluctuations in demand. Therefore, many companies are trying to produce products by introducing 3D printing technology into the manufacturing process. The 3D printing process is based on additive manufacturing (AM), which can fabricate complex shapes and reduce material waste and production time. Although AM has many advantages, its product quality is poor compared to conventional manufacturing systems. This study proposes a methodology to improve the quality of AM products based on data analysis. The targeted quality of AM is the surface roughness of the stacked wall. Surface roughness is one of the important quality indicators and can cause short product life and poor structure performance. To control the surface roughness, the resultant surface roughness needs to be predicted in advance depending on the process parameters. Various analysis methods such as data pre-processing and deep neural networks (DNN) combined with sensor data are used to predict surface roughness in the proposed methodology. The proposed methodology is applied to field data from operated wire + arc additive manufacturing (WAAM), and the analysis result shows its effectiveness, with a mean absolute percentage error (MAPE) of 1.93%.

## 1. Introduction

Additive manufacturing (AM) is a production method in which raw materials such as thermoplastics, ceramic powders, paper, plastic films, or metals are stacked layer by layer. Due to the characteristics of this process, AM has several advantages: (1) manufacturing products with complicated shapes; (2) producing small-quantity batches quickly; and (3) saving materials compared to subtractive manufacturing methods [1,2,3]. AM can be processed via various forms, such as extrusion, jetting, light polymerization, sintering, directed energy deposition (DED), lamination, and powder bed fusion (PBF) [4]. Among them, metal AM (i.e., DED, PBF) is attracting more attention since many machinery components should be produced with metal.

When focusing on metal AM, PBF uses high-energy power sources such as lasers or electron beams to melt or sinter material powder. The cost of the equipment and materials for PBF is high, and production speed is relatively low. In analogy to PBF, DED uses a focused energy source. However, in DED, the material is simultaneously melted as it is deposited by a nozzle, which helps to reduce material waste. One of the representative forms of DED is wire + arc additive manufacturing (WAAM), which uses metal wire as the feedstock and an arc as an energy source. Since the material in WAAM is deposited through a metal wire, the amount of metal used can be minimized. In addition, arc welding requires cheaper equipment than PBF or other DED methods. Despite its many advantages, WAAM is one of the lesser-known metal AM technologies. However, it has huge potential for large-scale metal AM applications across various industries.

Although WAAM has many benefits, most companies still hesitate to adopt it into their processes due to certain drawbacks. Additive processes based on arc welding can raise many problems, such as spatter, porosity, undercutting, deformation, cracks, and slag. In addition, the surface quality of layers stacked by WAAM is poor. The high heat energy of arc welding induces high residual stress and distortion, which deteriorates part accuracy and surface roughness. Therefore, additional post-processing, such as machining, is necessary, which leads to increased manufacturing costs.

To improve the quality of WAAM products and to reduce additional processing costs, it is important to control the surface roughness. Unlike the conventional definition of surface roughness used in the cutting process, surface roughness in WAAM is defined as the side of the wall built by stacked layers (see Figure 1). Since WAAM products are produced as stacked layers through welding, they can have harsh surface conditions compared to products produced by cutting, also known as the stair-stepping effect [5].

To reduce the additional finishing processes required for the surface roughness of stacked layers, additional layers must be deposited as flat as possible. The shape of the additionally stacked layers is decided by process parameters such as voltage, current, and feed rate [2,5,6,7]. In a case where a proper set of process parameters has been defined, the stair-stepping effect of the wall in WAAM can be reduced. In finding the near-optimal process parameters, it is necessary to predict the resultant surface roughness depending on the process parameters. This study proposes a methodology to predict surface roughness when an additional layer is stacked under a specific set of process parameters. The proposed method consists of data pre-processing to utilize raw data as input/output variables for the predictive model and implements machine learning algorithms such as DNN to predict surface roughness. Different statistical methods such as correlation analysis are also applied to verify the effectiveness of the prediction model. The usefulness and feasibility of the proposed methodology are proved by the experimental data collected from the gas metal arc welding (GMAW)-WAAM system. The remainder of the paper is organized as follows: Section 2 reviews the related work to analyze the influence of process parameters on surface roughness and on predictions of surface roughness. Section 3 and Section 4 presents the data analysis-based predictive modelling approach to predict surface roughness, and Section 5 provides conclusions and directions for future work.

## 2. State of the Art

This section discusses the previous works related to surface roughness measurement and the process parameters affecting surface roughness during AM processes as well as some prediction models for surface roughness.

### 2.1. Definition of Surface Roughness

In general, surface roughness is defined as regular or irregular unevenness on a surface. Product surface embodies a complex microshape made of a series of peaks and troughs of varying heights, depths, and spacings. In the case of large components, the effect of surface roughness can be negligible since it only affects small areas within micron ranges. However, surface roughness is more critical as more components are being miniaturized [8]. Therefore, many research works have tried to define surface roughness precisely and to control it during manufacturing.

Surface roughness can be measured by various methods depending on different definitions [9,10,11,12,13]. The mostly used measurement method for surface roughness is the average distance between the surface and mean surface profiles (see Figure 2). As shown in Figure 2, the shaded area should be summed and divided by the length L to calculate the surface roughness.

Lee [9] suggested surface waviness by calculating the effective area (EA) ratio from the cross-section of an orthogonally cut wall. The EA ratio is calculated by dividing the whole area of the cross-section of a wall by the area of the largest inscribed rectangle. Since surface waviness is calculated from one cross-section of the wall, it is challenging for it to represent the general surface roughness of the whole side of the wall. To overcome this limitation, some research works have tried to obtain as much data from the wall surface as possible. Some researchers [10,11] measured surface roughness by calculating the distances between the surface profile points and the above mean plane defined by the surface profile. Since more wall points are considered when calculated using 3D scanner data to measure surface roughness, the surface roughness is more accurate. Since the surfaces of the walls in WAAM products are highly uneven, considering more points when measuring surface roughness is desirable. Therefore, this study uses cloud points from the surface measured by a 3D scanner.

### 2.2. Process Parameters Related to Surface Roughness

Chan et al. [14] conducted a study regarding the effect of surface roughness on product life. According to this study, surface roughness causes a reduction in product life expectancy. Sahin [15] explained the effects of surface roughness on product performance, such as tensile strength and fatigue strength. Dawood et al. [11] analyzed the influence of surface roughness on microstructures and mechanical properties. According to these works, surface roughness is one of the most important factors of product quality, and it should be controlled carefully. Considering WAAM, the surface roughness becomes more prevalent at a macroscale, so the fineness of the walls created by WAAM is not as important. The important aspect of surface roughness in WAAM is that the uneven surface caused by the stair-stepping effect requires more post-processing. However, more post-processing cannot guarantee sufficient wall thickness. 

Some research works are dedicated to finding the relation between the control variables of processing and surface roughness. Xiong et al. [5] studied the influence of process parameters on surface roughness in the case of gas metal arc welding (GMAW). In this study, process parameters such as the inter-layer temperature, wire feed speed, and travel speed are shown to be closely related to surface roughness. Galantucci et al. [16] analyzed the effect of the process parameters on surface roughness in fused filament fabrication (FDM)-based AM via the design of the experiments. This study proved that the surface roughness deteriorates with increasing slice height and raster width. In other work, the five shape measurements of beads (layer thickness, build angle, raster angle, raster width, and air gap) seem to be essential variables to change surface roughness [17]. Strano et al. [18] studied the effect of layer thickness on surface roughness for steel 316 alloy parts made by selective laser melting (SLM). Zhou et al. [1] chose four important parameters (layer thickness, printing saturation, heater power ratio, and drying time) when developing a prediction model for surface roughness since those are highly effective. Yamaguchi et al. [19] studied the effect of heat input and argon gas on surface roughness. This study showed that increasing the heat input deteriorates the surface roughness and that argon gas helps surface roughness more than other shield gases. Bhushan and Sharma [6] investigated the impact of welding factors such as rotational speed and welding speed regarding the surface roughness of friction stir-welded AA6061-T651. Their results showed that the rotational speed of 1400 rpm and the welding speed of 20 mm/min resulted in the finest surface roughness. Chinchanikar et al. [7] carried out an investigation regarding the effects of different combinations of process parameters (rotational speed and feed rate) on surface roughness when welding aluminum 6063 alloy. Dinovitzer et al. [2] analyzed the influence of travel speed and current on surface roughness. According to this analysis, increasing travel speed and decreasing current worsen the surface roughness. From the previous works, it is proven that various process parameters effect surface roughness, and this study focuses on two of them (i.e., feed rate and travel speed) as control variables.

### 2.3. Prediction of Surface Roughness

Swarna and Arumaikkannu [20] proposed a non-contact method for estimating the surface roughness in SLM-customized implants using an artificial neural network (ANN). The ANN developed in the study was used to predict surface roughness after training using scan data from a femur bone. The prediction accuracy reached 97.2%. Ahn et al. [21] developed a prediction model to estimate the surface roughness of a whole area and the distribution of the surface roughness in a sampled area using interpolation. Strano et al. [18] predicted the surface roughness by considering the stair-stepping effect for SLM specimens, thus helping to minimize the need for post-processing. Boschetto et al. [22] developed an ANN model to determine the surface roughness of FDM parts. The model is used to optimize the effect of process parameters in the product development stage. Wu et al. [23] proposed a data fusion approach to predict surface roughness in FDM processes. This study combines three kinds of sensor data (vibration, temperature of the extruder and table, melt-pool temperature) with various artificial intelligence (AI) models. Vahabli and Rahmait [17] also used an ANN model to predict surface roughness. Chen and Zhao [24] adopted a backward propagation neural network (BP-NN) to predict surface roughness. Xia et al. [25] developed a prediction model to predict the surface roughness in WAAM processes. This study calculated the surface roughness using a laser scanner and combines three kinds of parameters (welding speed, wire feed speed, and overlap ratio) with a genetic algorithm–adaptive neuro fuzzy inference system (GA-ANFIS). The prediction model shows its performance with a MAPE of 14.15%. Yaseer and Chen [26] investigated the layer roughness in WAAM processes. This study explored a layer-roughness prediction method based on multilayer perceptron (MLP) and random forest combined with weaving path. Their results show that random forest achieved better performance in terms of MAPE, the value of which is about 5.64%.

As described in the previous literature, many studies have tried to predict surface roughness depending on process parameters and using various methods in the finishing process. Recently, some researchers have tried to predict surface roughness in situ; however, to our knowledge, using previously stacked layers as input data has not been considered. Studies predicting the surface roughness of stacked layers in situ during WAAM are still lacking. Therefore, this study focuses on developing a prediction model for surface roughness in situ during WAAM by considering process parameters and previously stacked layers using an AI model.

## 3. Prediction Model of Surface Roughness

The stair-stepping effect can be minimized by properly setting process parameters such as current, voltage, and feed rate. However, there are numerous possible combinations that only use three parameters. In finding the near-optimal ones, it is necessary to be able to predict the surface quality depending on a set of process parameters. Then, each set should be assessed and compared to search for the best process setting. This section will show how the surface roughness can be predicted depending on various process parameter settings. Finding the best set of process parameters among many candidates will be carried out in future work.

### 3.1. Measurement of Surface Roughness

The first step in predicting surface roughness is to define it precisely. As previously described, the surfaces of walls made via WAAM are harsher than those of a conventionally manufactured (i.e., cutting, drilling, punching, etc.) product. However, the basic concept of measuring the surface roughness is same. The important difference from the conventional definition of surface roughness is that the surface roughness of the wall in the WAAM product is generated layer by layer at the macrolevel. Since the surface roughness is redefined whenever a new layer is stacked in the WAMM process, the surface roughness in this study is defined between two consecutive stacked layers, as shown in Figure 3.

The method used for measuring surface roughness is depicted in Figure 4. The main concept of defining surface roughness is to measure the variation in the surface profile between consecutively stacked layers. The starting and ending point between two layers is set to the widest area of each layer when the WAAM wall is cut orthogonally. The actual profile is obtained using a coordinate measuring machine (CMM, model: Hexagon Romer Arm 7525SIE) as points clouds (see Figure 4b).

Generally, the surface roughness is measured using one cross-section cut from the wall (See Figure 3 red in B and Figure 4 red in B). So that it is limited to representing the whole surface area of the wall (See Figure 4). In this study, the surface profile measured by CMM is expressed as three-dimensional coordinates ((*x_k,L_,y_k,L_,z_k,L_*) for the left-wall side and (*x_k,R_,y_k,R_,z_k,R_*) for the right-wall side) of the point clouds extended from one cross-section and can include the characteristics of the whole surface area of the wall (see Figure 3a and Figure 4b). The numerical equation to calculate the surface roughness between two layers, which considers the wall’s whole surface area, is formulated as Equation (1).
(1)$$\mathrm{Surface}\;\mathrm{rouhgness}=(\frac{\frac{\sum_{k=1}^{n}{\left(y_{k,L}-\bar{y_{L}}\right)}^{2}}{n}+\frac{\sum_{k=1}^{m}{\left(y_{k,R}-\bar{y_{R}}\right)}^{2}}{m}}{2}$$
where $y_{k,L}$ is the observed *y* coordinate value of the *kth* point cloud on the left surface profile of the wall. The index of *k* ranges from 1 to *n* since there are *n* points on the left side of the wall. Additionally, $\bar{y_{L}}$ is the mean of the *y*-values for the point clouds on the left surface. On the other hand, $y_{k,R}$ is the observed y-values of the right-side surface profile of the wall. The point clouds on the right side of the wall consist of *m* points. The deviation in the left side of the wall is calculated by subtracting the observed $y_{k,L}$ value of each point from the mean ($\bar{y_{L}}$) and then squaring and adding all of them and then dividing by the total number of observed point clouds from the surface profile on the left. The deviation in the right side of the wall also follows the same method. The surface roughness is measured as the mean of the deviation in both sides since the manufactured wall is composed of a left and right side.

### 3.2. Experimental Set-Up

The experimental setup in this study is based on gas tungsten arc welding–cold metal transfer (GMAW)-CMT, as shown in Figure 5. This system consists of a robot manipulator (Fanuc ArcMate 120iC) and a welding power source (Fronius TPS 400i) equipped with a welding torch (Fronius WF 25i Robacta Drive). The process parameters were controlled using the robot and power-source controllers. A coordinate measuring machine (CMM) was also installed to obtain 3D point clouds of the wall surface between two stacked layers.

A wall made by (GMAW)-CMT was built on a stainless steel 316 L substrate with the dimensions 6 × 2 × 0.25 inches. A bead deposition experiment was conducted on the substrate using stainless steel 316 L as a wire material, and the deposition direction was unaltered. Both middle sides of the substrate were clamped to the stage to prevent distortion, and wire was fed at an angle of 30° from the top surface towards inside.

Since the welding is processed by a robot arm, there can be positioning accuracy problems. The wall roughness should be controlled at the macrolevel in this work. Therefore, processing errors caused by robot arm movement are not considered in this paper.

### 3.3. Data Collection and Pre-Processing

#### 3.3.1. Process Parameters

The process parameters of (GMAW)-CMT deposition are shown in Table 1. The dynamic (or controlling) process parameters are newly set whenever the next layer is deposited. The static parameters are fixed until the process is finished. The travel speed can vary from 1 to 12,000 cm/min, with one-unit increments. The feed rate can be set from 100 to 1000 cm/min and changes in increments of 10 units. The deposited layer is cooled to 100 °C before the next layer is stacked to reduce the influence of heat.

#### 3.3.2. Bead Shape

As mentioned in Section 2.2, process parameters (travel speed and feed rate) highly influence the surface roughness. In addition, the bead shape (angle, width, height, and bead location (layer)) of the previous layer (see Figure 6) also plays an important role in shaping newly stacked layers, deciding the surface roughness. Therefore, this study considers the bead shape and dynamic process parameters in Table 1 as the input data to develop an AI model for predicting the surface roughness. According to previous studies, the hardness of the bottom layer deposited using a WAAM process is not constant due to mechanical properties. Therefore, the substrate and bottom layer are not used at actual worksites [27,28,29]. Thus, only the data for the beads deposited above the first layer of the wall are used in model development (Figure 6).

In Figure 6, W represents the width, which is the widest distance between each layer. Each layer’s height (H) is the distance from the top of the previous layer to the top of the next layer. The angles for both sides ($\theta_{Left},{\;\theta }_{Right}$) of each layer are measured based on the narrowest area between two stacked layers.

#### 3.3.3. Data Collection

To collect experimental data regarding dynamic process parameters and bead shape, 27 thin walls with five layers were fabricated using the (GMAW)-CMT system. The three combinations of dynamic process parameters shown in Table 2 are set to deposit the layers.

The process for collecting and pre-processing data such as the dynamic process parameters and bead shape is shown in Figure 7.

Generally, the bead shape is measured by one cross-section of the wall so that it has a limited capability in representing the whole-wall characteristics. To improve this limitation, point clouds of the surface profile are cut into 50 cross-sections at regular intervals for each wall. The surface profile of each cross-section composed of point clouds is converted to one line using the smoothing method (see lower image of Figure 7). Then, using each cross-section, the bead shape is measured as described in Section 3.3.2. The mean of the bead shape of all cross-sections represents the bead shape of the whole area of each wall. Next, the input and output data used to train the model are defined by the measured bead shape, dynamic process parameters, and surface roughness.

### 3.4. Model Development

#### 3.4.1. Definition of Input Data and Output Data and Normalization

To develop an AI model, defining the input and output data is a prerequisite. The collected data (explained in Section 3.3), such as the dynamic process parameters and bead shape of the previously stacked layer, are adopted as the features of the input layer of the AI model, and one resulting property (surface roughness between the consecutively stacked layers of a wall) is adopted as the output-layer variable (refer Table 3). Collected structural data, samples of dynamic process parameters, bead shape, and surface roughness obtained from the experiments are described in Table 3.

The first two columns in Table 3: ‘Index’ and ‘Number of thin walls’, represent an index of the two consecutive layers processed in each wall. In each wall, there are four consecutive layers. The third column indicates the data measured from the processed layer in the previous deposition, and the bead shape is expressed from the fourth column to the seventh. The eighth and ninth columns show the dynamic process parameters for the layer currently being processed. The last column is for the output value of the surface roughness between the previous layer and the layer currently being processed under the given process parameter are in (the eighth and ninth columns).

Normalization is performed for each variable since the measured values have different value ranges. The aim is to reduce the influence of the deviations caused by the differences in the measurement range of each variable. In addition, normalization can reduce the learning time of machine learning models and prevent decreases in accuracy caused by heavy computations [30]. In this study, the robust scaler normalization is used, where it is represented in Table 4.

#### 3.4.2. DNN Model Development

There are various kinds of machine learning models. ANN is a single-layer perceptron structure, which has limitations in solving nonlinear problems. To cover this limitation, a deep learning model with multiple hidden layers using backpropagation is proposed. Recently, DNN has been widely used in various areas and has shown good performance. Hence, the authors adopted a DNN-based model. DNN is one of the core models of deep learning and has a structure comprising multiple hidden layers. It has the advantages of understanding the complex structure of large datasets and learning various non-linear relationships. The proposed structure is shown in Figure 8. The structure of DNN can vary according to the hyper parameters (number of hidden layers, optimizer, learning rate). Therefore, finding the best structure for DNN requires trial and error.

## 4. Model Evaluation

To evaluate the prediction performance of the used DNN model, mean absolute percentage error (MAPE) and root mean squared error (RMSE) are adopted as performance measures (refer to Equations (2) and (3)). Since MAPE represents the averaged difference in the percentage between real and predicted values, a lower value means better performance. The range of MAPE ranges from 0 to 100. RMSE is used to represent the precision of the model, and a lower value means the better precision. The range of RMSE become from 0 to ∞. The formulae for MAPE and RMSE are as expressed in Equations (2) and (3).
(2)$$\mathrm{MAPE}=\frac{100}{n}\overset{\;\;n}{\underset{t=1}{\sum }}\left|\frac{A_{t}-F_{t}}{A_{t}}\right|$$
(3)$$\mathrm{RMSE}=\sqrt{\overset{\;\;n}{\underset{t=1}{\sum }}\frac{{\left(A_{t}-F_{t}\right)}^{2}}{n}}$$

In Equations (2) and (3), *A_t_* represents the actual surface roughness calculated by CMM data, and *F_t_* is the surface roughness predicted by the used model. The whole data set is split into 80% training data and 20% testing data. The training data set is only used to learn the model, and the testing data set is used to see how well the model performs under the new process parameter settings. Table 5 shows the results of the performance measures depending on the tested prediction models. Some conventional predictive models, such as regression and support vector regression (SVR), are also tested to compare them with DNN.

From Table 5, the prediction accuracy of the polynomial (quadratic) regression model gives the best results, with an MAPE of 7.75 % and an RMSE of 0.126. As a result, linear regression is not enough to explain the data set, and polynomial (cubic) regression is over-fit on the training data set. SVR does not achieve large variance of accuracy depending on the hyper parameters of the model. However, the error rate of SVR has an MAPE of 8.31%, but that it still is not enough to explain the data set. On the other hand, the prediction accuracy of the DNN model with the following hyper parameters: Activation function = Relu; Layer = (input layer (7), hidden layer (64, 32, 16), and output layer (1); value in () means number of nodes of each layer); Drop out = None; Weight initialization = He initialization; Optimizer = Adam; Learning_rate = 1 × 10^−4^; and Epoch = 15,000, gives the best result, with MAPE = 1.93% and RMSE = 0.03. Figure 9a shows the residual error between the actual surface roughness and the predicted surface roughness as a graph. To validate the prediction accuracy of the second DNN model with the highest one, the correlation between the real value and the value predicted by the DNN model is plotted. Correlation analysis is commonly used to infer the relationship between two variables. The value of the Pearson correlation shows a positive correlation when the value of each variable increases or decreases together. When the value of one variable increases and the value of the other variable decreases, it shows a negative correlation. When the value of one variable change and the value of one variable remains the same, it means there is no correlation between the two variables. The correlation coefficient, ‘r’, always satisfies −1 ≤ r ≤ 1. In the case of no correlation, the value of r is close to 0. When r is more than 0.6 or less than −0.6, the correlation between the variables is strong. In the DNN model used in this study, the correlation coefficient between the actual and predicted value is r = 0.97, which means that they have a strong correlation. Figure 9b shows the relationship between the actual surface roughness and the predicted surface roughness as a graph.

## 5. Conclusions and Discussion

Concerning the comparison with related work, the proposed method is able to predict surface roughness in situ. If the surface roughness can be reduced in situ by predicting the surface roughness depending on the process parameters, there will be less post-processing, and material waste will be reduced. To achieve this, some studies have proposed AI models based on process parameters. Some of the researchers who have tried to predict surface roughness in situ have used process parameters (welding speed, wire feed speed, and overlap ratio). However, there have been few investigations of the bead geometry of previously stacked layers, which has a huge potential to reduce surface roughness. Additionally, researchers have previously defined surface roughness using the top surface of a bead. However, the surface roughness of the side of a bead also requires post-processing in multi-layer processes.

Therefore, this paper proposed a prediction model for surface roughness between consecutively stacked layers in a thin wall produced via WAAM that depended on process parameters and the bead shape of the previously processed layer using DNN. The targeted WAAM process was a (GMAW)-CMT system that could monitor and collect dynamic process parameters. Two kinds of dynamic parameters (travel speed and feed rate) are focused on the (GMAW)-CMT process. The bead shape was measured by the CMM that was installed and maintained a 3D position as point clouds in a thin wall. These data are used as input data in the used DNN model, which can predict surface roughness under given process parameters. Commonly, when measuring surface roughness, one cross-sectional of bead is used. This has limitations in representing the whole area of a wall. To cover this limitation, we propose an extension of the cross section to the whole area of the wall using point clouds with a smoothing method. In addition, robust scaler was adopted to reduce the influence of the deviations caused by the difference in the measurement range of each variable during the analysis.

Some conventional predictive models such as regression and SVR were also adopted, and the prediction performances were compared with the used DNN model. According to the experimental results, the DNN model showed the best performance among them. The best DNN model has predication accuracy of about 98%, with a high correlation between the real and predicted values. Using the developed model, surface roughness can be estimated when a new layer is stacked under a diverse combination of bead shapes and dynamic process parameters. In the process results achieved under a specific operation parameter, the best operation parameter can be searched. Then, a search algorithm or reinforced learning should be adopted. This will be carried out in other authors’ future research.

## Figures and Tables

**Figure 1 sensors-22-07955-f001:**
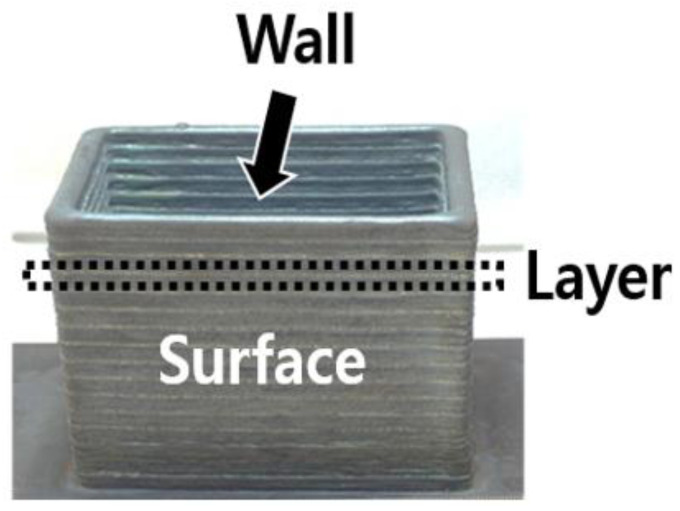
Surface roughness of WAAM product.

**Figure 2 sensors-22-07955-f002:**
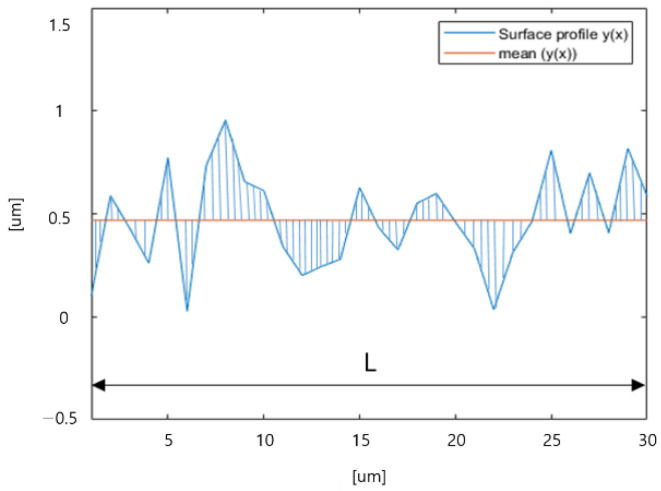
Calculation of surface roughness using surface profile.

**Figure 3 sensors-22-07955-f003:**
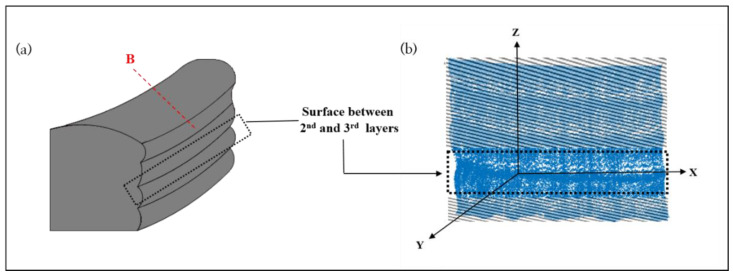
(**a**) Surface of WAAM product between second and third layers. (**b**) Reconstruction of WAAM product between second and third layers by CMM.

**Figure 4 sensors-22-07955-f004:**
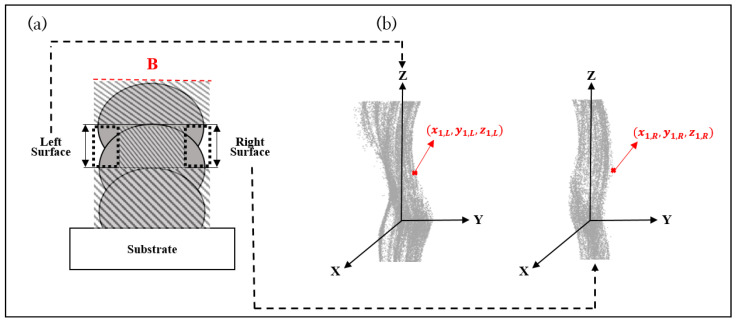
(**a**) Schematic diagram of cross-section of WAAM product. (**b**) Reconstruction of surface profile of both sides of a WAAM product between second and third layers by CMM.

**Figure 5 sensors-22-07955-f005:**
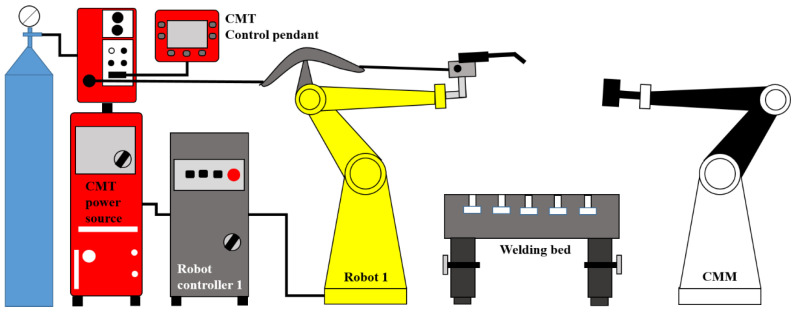
Environment of the experimental set-up.

**Figure 6 sensors-22-07955-f006:**
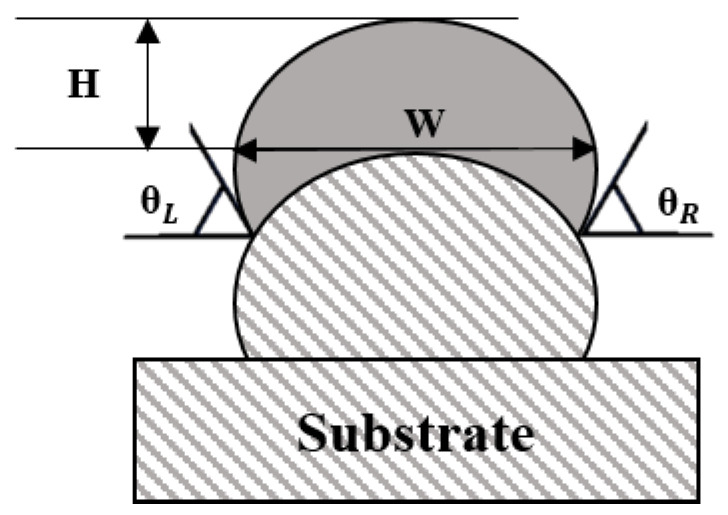
Schematic diagram of bead shape.

**Figure 7 sensors-22-07955-f007:**
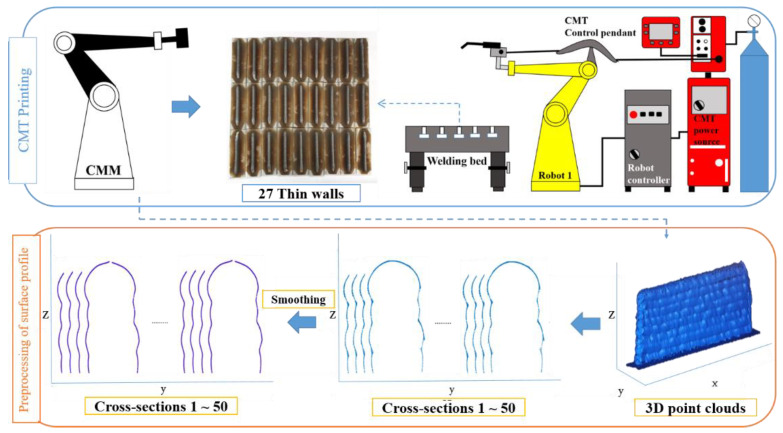
Pre-process and process for data collection.

**Figure 8 sensors-22-07955-f008:**
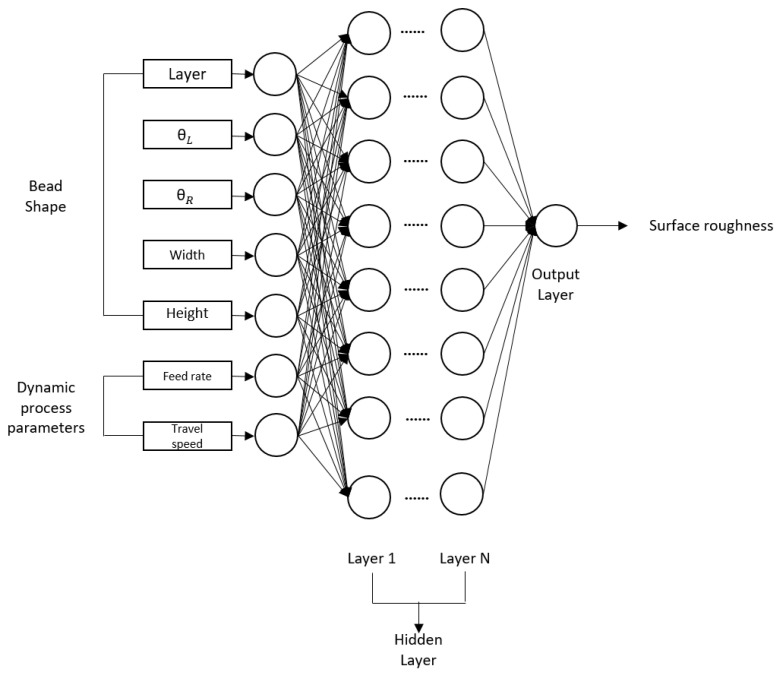
Schematic diagram of the deep neural network for the prediction of surface roughness.

**Figure 9 sensors-22-07955-f009:**
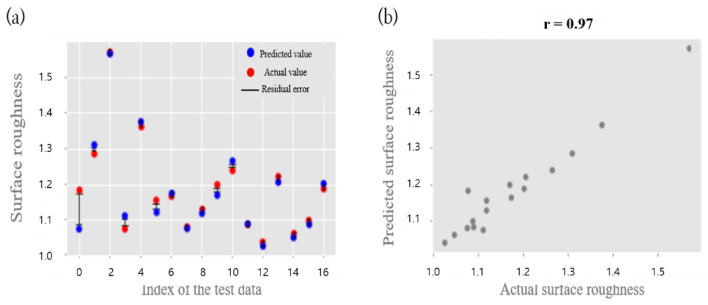
(**a**) Residual error plot of real values and predicted values. (**b**) Scatter plot of real values and predicted values.

**Table 1 sensors-22-07955-t001:** WAAM process parameters.

Parameters		Unit	Values
Dynamic process parameters	Travel speed	cm/min	1~12,000
	Feed rate	cm/min	100~1000
Static process parameters	Previous layer temperature	°C	100
	Arc length (bead to arc distance)	mm	5
	Wire diameter	mm	1.2
	Wire feeding angle	Degree	30
	Shielding gas	%	100
	Flow rate	L/min	20

**Table 2 sensors-22-07955-t002:** WAAM dynamic process parameters.

Combination No.	Feed Rate	Travel Speed
1	480	30
2	560	31
3	650	33

**Table 3 sensors-22-07955-t003:** Input and output structure.

		Input Data							Output Data
Index	# of Thin Wall	Layer	$\theta_{L}$ (°)	$\theta_{R}$ (°)	Width (mm)	Height (mm)	Travel Speed (cm/min)	Feed Rate (cm/min)	Surface Roughness
1	1	2nd	102.11	101.94	6.95	2.06	30	480	1.0363
2	1	3rd	100.98	99.87	6.91	3.73	31	560	1.1303
︙	︙	︙	︙	︙	︙	︙	︙	︙	︙
80	27	3rd	99.37	100.2	7.36	2.76	31	560	1.0393
81	27	4th	100.05	99.29	7.46	2.76	31	560	1.0474

**Table 4 sensors-22-07955-t004:** Data normalization.

		Input Data							Output Data
Index	# of Thin Wall	Layer	$\theta_{L}$ (°)	$\theta_{R}$ (°)	Width (mm)	Height (mm)	Travel Speed (cm/min)	Feed Rate (cm/min)	Surface Roughness
1	1	−0.5	−0.69	−3.19	0.64	0.65	−1	−8	1.0363
2	1	0	−0.76	3.4	0.23	0.05	0	0	1.1303
︙	︙	︙	︙	︙	︙	︙	︙	︙	︙
80	27	0	−0.14	−0.4	−0.35	0.14	0	0	1.0393
81	27	0.5	0	−0.4	−0.1	−0.12	0	0	1.0474

**Table 5 sensors-22-07955-t005:** Results of performance comparison.

Model	Model Parameters							Result	
Regression	Degree							Mape (%)	Rmse
1	Linear							67.33	0.97
2	Polynomial(quadratic)							7.75	0.13
3	Polynomial(cubic)							18.08	0.3
**SVR**	**C**		**Degree**		**Epsilon**	**Kernel**		**Mape (%)**	**Rmse**
1	1		3		0.1	rbf		8.31	0.109
2	1.3		2		0.1	Poly		8.43	0.111
3	1.5		4		0.1	sigmoid		12.2	0.183
**DNN**	**Activation Function**	**Layer**	**Drop Out**	**Weight Initialization**	**Optimizer**	**Learning Rate**	**Epoch**	**Mape (%)**	**Rmse**
1	Relu	(7, 64, 32, 28, 1)	0.2	He initialization	RMSprop	1 × 10^−2^	10,000	10.45	0.145
2	Relu	(7, 64, 32, 16, 1)	None	He initialization	Adam	1 × 10^−4^	15,000	1.93	0.03
3	Relu	(7, 32, 16, 1)	0.3	He initialization	SGD	1 × 10^−3^	30,000	11.39	0.16

## Data Availability

Not applicable.

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
