# Peer review of "Prediction of Metal Additively Manufactured Surface Roughness Using Deep Neural Network"

_sensors, 2022, doi:10.3390/s22207955_

Round 1
Reviewer 1 Report
This paper has a good theoretical basis and application value for the detection of surface roughness. There are still some details that need to be further improved. First, in the results and discussion section, the number of samples used for deep learning model detection should be given, as everyone knows, deep learning algorithms have a great demand for the number of samples, and the sample size given in Figure 9 is too small to illustrate the reliability of the detection results. Second, the experimental device in Figure 5 should clarify the positioning error problem of the mechanical arm, and in high-precision detection, the positioning accuracy is too low, which will have a great impact on the detection result of roughness, especially in the application scenario of um level. Third, the English writing aspect needs to be further revised.
Author Response
Review's comment |
This paper has a good theoretical basis and application value for the detection of surface roughness. There are still some details that need to be further improved. First, in the results and discussion section, the number of samples used for deep learning model detection should be given, as everyone knows, deep learning algorithms have a great demand for the number of samples, and the sample size given in Figure 9 is too small to illustrate the reliability of the detection results. Second, the experimental device in Figure 5 should clarify the positioning error problem of the mechanical arm, and in high-precision detection, the positioning accuracy is too low, which will have a great impact on the detection result of roughness, especially in the application scenario of um level. Third, the English writing aspect needs to be further revised. |
Thank you for your valuable time on reviewing the proposed paper. I appreciate the effort that you dedicated to providing feedback on our manuscript and are grateful for the insightful comments on and valuable improvements to our paper. I have incorporated most of the suggestions made by your comments. Those changes are highlighted within the manuscript in red. Please see below point-by-point the response to your comments and concerns.
|
(1) In the results and discussion section, the number of samples used for deep learning model detection should be given, as everyone knows, deep learning algorithms have a great demand for the number of samples, and the sample size given in Figure 9 is too small to illustrate the reliability of the detection results. |
-As reviewer’s comments, the larger dataset guarantees the more reliability on the model. We have tried to collect as much as data. However, WAAM requires lots of setting time for each parameter set up and we need various combination of parameters, which makes limitation on data acquisition. To relieve this limitation, the controllable parameters are decided to be limited as two kinds (dynamic process parameters such as travel speed and feed rate) with less combination to reduce data dimension space so that we hope the used model can give reasonable result. In the given data, the training data and test data are separated and the DNN model showed good prediction for the test data. Hence, we expect the used DNN model can work well in the given data. Moreover, we collected data based on design of experiment so that the possible data dimension area of data can be covered with less data. Even though the tested data is short, we expect that the DNN model can give generalization of other process parameters.
|
(2) The experimental device in Figure 5 should clarify the positioning error problem of the mechanical arm, and in high-precision detection, the positioning accuracy is too low, which will have a great impact on the detection result of roughness, especially in the application scenario of um level. |
- As reviewer commented, the positioning error can happen with WAAM process. However, the wall product made by WAAM requires post-process in making fine surface. The objective of measuring surface roughness in WAAM is to reduce as much as removed material during surface grinding. Too rough surface needs much grinding, which cannot guarantee enough thickness of wall after surface grinding. Therefore, in this work, the um level controlling is not much important. For the readers, the following sentence is added in line 247-250. “Since the welding is process by robot arm, there can exist the positioning accuracy problem. The roughness of wall can be controlled by mm level in this work and the postprocess should be added. Therefore, the processing error by robot arm is not considered in this paper.”
|
(3) Third, the English writing aspect needs to be further revised. |
- As you commented, we have checked and modified English errors in the paper.
|

Reviewer 2 Report
- The lack of a computer science and AI is evident and strobngly felt in this paper. The paper is about estimating surface roughness of the 3D manufacturing process.
The industrial engineering part is eclipsing other computer science and AI parts. More emphasis should be made on deep learning models and less on the generation of data. Moreover, there is no mention of why deep learning models are actually needed in comparison to traditional neural networks.
- A good contribution would have been to estimate surface roughness from images, and this case, DNN would be a great fit.
- Table 5 need to be succinct with model details moved into other parts.
- Do the results present testing or validation, because there is no mention of such details.
- Compare the results from related works.
- Don't include undefined abbreviations in the abstract (e.g., MAPE)
- improper sentence, line 20.
- There is no list of abbreviations.
- multiple references should be included in the same brackets (e.g., [1,2,3]), line 35.
- line 49, provide reference, similarly, lines 51,52.
- Make sur the template is followed properly (e.g., Fig. 1 not Figure 1).
- Sction 2.3, don't start the paragraph with a reference number. This also applies to other sentences, do not use the reference number as subject (e.g., [2] analyzed --> Dinovitzer et al. [2] analyzed ).
- SVR is not defined.
Author Response
Review comments |
|
Thank you for your valuable time on reviewing the proposed paper. I appreciate the effort that you dedicated to providing feedback on our manuscript and are grateful for the insightful comments on and valuable improvements to our paper. I have incorporated most of the suggestions made by your comments. Those changes are highlighted within the manuscript as red. Please see below, in red, for a point-by-point response to your comments and concerns.
The lack of a computer science and AI is evident and strongly felt in this paper. The paper is about estimating surface roughness of the 3D manufacturing process. (1) The industrial engineering part is eclipsing other computer science and AI parts. More emphasis should be made on deep learning models and less on the generation of data. Moreover, there is no mention of why deep learning models are actually needed in comparison to traditional neural networks. - As you commented, we supplement the reason for using DNN as below. (see lines 325-328) “ANN is a single-layer perceptron structure, which is limited in solving nonlinear problems. To do this, a deep learning model with hidden layers has been proposed using backpropagation” -To validate the use of DNN model, we tested conventionally used prediction model such as regression or support vector model. In this comparison, the DNN gives the better result A good contribution would have been to estimate surface roughness from images, and this case, DNN would be a great fit. (2) Table 5 need to be succinct with model details moved into other parts. - As you commented, we modified Table 5 as succinct. (3) Do the results present testing or validation, because there is no mention of such details. - It can be shown that whole data set split into 80% as train data set and 20% as test data set. (see in lines 346~349) (4) Compare the results from related works. - Thank you for kind comments, as suggested, we enriched the related works to compare the result from related works (see in lines 170-178). ‘Xia et al. [29] developed a prediction model to predict the surface roughness in WAAM processes. This study calculated the surface roughness by a laser scanner and combines three kinds of parameters (welding speed, wire feed speed and overlap ratio) with genetic algorithm-adaptive neuro fuzzy inference system (GA-ANFIS). The prediction model shows its performance with MAPE of 14.15%. Yaseer and chen [30] investigated the layer roughness in WAAM process. This study explored layer roughness prediction method based on multilayer perceptron (MLP) and random forest combined with weaving path. Their result shows that random forest achieved better performance in terms of MAPE which is about 5.64%.’ - Also, we have added more information, in order to claim the unique and solid contributions of this paper (see in lines 180-185 and 382-391). “As described in the previous literature, many studies tried to predict surface rough-ness depending on process parameters with various methods in finishing process. Recently, some researchers have tried to predict surface roughness in situ, however, to our knowledge, previously stacked layer as input data has not been considered. The study on predicting surface roughness of the stacked layers in situ of WAAM is still lacking. Therefore, this study focuses on developing the prediction model for surface roughness in situ of WAAM considering process parameters and previously stacked layer using AI model.” ‘Concerning the comparison with related work, the proposed method able to predict surface roughness in situ. If the surface roughness can be reduced in situ by predicting surface roughness depending on the process parameters, less post processing and material wastage will be reduced. To do this, some studies proposed AI model based on the process parameters. some researchers who tried to predict surface roughness in situ used process parameters (welding speed, wire feed speed and overlap ratio). However, the investigation of the bead geometry of previously stacked layers, which has a huge potential to reduce surface roughness, has not been considered well. Also, researchers defined surface roughness using the top surface of the bead. However, surface roughness of side of the bead is also a part that requires post-processing in multi-layer process.’ (5) Don't include undefined abbreviations in the abstract (e.g., MAPE) - As you commented, we modified definition of MAPE (see in line 27). (6) improper sentence, line 20. - The sentence is modified as follows (see in line 19-22). This study proposes a methodology to improve the quality of AM products based on data analysis. The targeting quality of AM is the surface roughness of the stacked wall. The surface roughness is one of the important quality indicators, which can cause low product life and structure performance. (7) There is no list of abbreviations. - The list of abbreviations is added in the last part of paper. (8) multiple references should be included in the same brackets (e.g., [1,2,3]), line 35. - As you commented, we modified reference numbers. (9) line 49, provide reference, similarly, lines 51,52. - As you commented, we modified multiple reference numbers according to the author guideline. (10) Make sure the template is followed properly (e.g., Fig. 1 not Figure 1). - As you commented, we modified “Fig” to “Figure” in the figures. (11) Section 2.3, don't start the paragraph with a reference number. This also applies to other sentences, do not use the reference number as subject (e.g., [2] analyzed --> Dinovitzer et al. [2] analyzed ). - As you commented, we modified references. (12) SVR is not defined. - As you commented, we defined the abbreviation of SVR (See in lines 350-351) |
Round 2
Reviewer 2 Report
- The authors addressed most of my comments. However, they should have checked their latest manuscript version before submission. The document should have been submitted without their personal notes.
- Again, I recommend the authors to include computer specialists in their future work. As hard they are trying to compensate for it, some little issues give away the lack of specialist knowledge in machine learning.
Author Response
2nd Review |
|
Thank you for your valuable time on reviewing the proposed paper. I appreciate the effort that you dedicated to providing feedback on our manuscript and are grateful for the insightful comments on and valuable improvements to our paper.
(1) The authors addressed most of my comments. However, they should have checked their latest manuscript version before submission. The document should have been submitted without their personal notes. - We really appreciate your kind comment. We did mistake to include our revision note inside of the paper. We have checked the modified parts and delete not necessary memo. (2) Again, I recommend the authors to include computer specialists in their future work. As hard they are trying to compensate for it, some little issues give away the lack of specialist knowledge in machine learning. - As you suggested, we will cooperate with machine learning specialist in our next work and will review our work for the high quality.
|